# The Influence of Fly Ash Content on the Compressive Strength of Cemented Sand and Gravel Material

Qihui Chai [1,2], Fang Wan [1,2,*], Lingfeng Xiao [1] and Feng Wu [1]

[1] School of Water Resources, North China University of Water Resources and Electric Power, Zhengzhou 450046, China; chaiqihui@ncwu.edu.cn (Q.C.); xlf19980616@163.com (L.X.); wufeng@ncwu.edu.cn (F.W.)

[2] Henan Key Laboratory of Water Resources Conservation and Intensive Utilization in the Yellow River Basin, Zhengzhou 450046, China

\* Correspondence: wanxf1023@163.com

**Abstract:** Cemented sand and gravel (CSG) material is a new type of dam material developed on the basis of roller compacted concrete, hardfill, and ultra-poor cementing materials. Its main feature is a wide range of sources of aggregate (aggregate is not screened but by simply removing the large particles it can be fully graded on the dam filling) and low amounts of cementitious materials per unit volume. This dam construction material is not only economical and practical, but also green and environmentally friendly. There are many factors affecting the mechanical properties of CSG materials, such as aggregate gradation, sand ratio, water content, water–binder ratio, fly ash content, admixture content, etc. Based on the existing research results of the team, this paper focuses on the influence of fly ash content on the compressive strength of CSG materials. Through a large number of laboratory measured data, we found: (1) The compressive strength law of materials at different ages; the compressive strength of CSG material at age 90 d is generally 10%~30% higher than that at 28 d, and it is proposed that 90 d or 180 d strength should be used as the design strength in the design of CSG material dam; (2) There is an optimal value of fly ash content in CSG materials: when the fly ash content is 50% of the total amount of cementitious materials (cement + fly ash), the fly ash content is defined as the optimal content, and the test data are verified by regression analysis. The discovery of an 'optimal dosage' of fly ash provides an important reference for the design and construction of CSG dams.

**Keywords:** cementitious gravel; fly ash; age; optimal dosage

## 1. Introduction

Cemented sand and gravel (CSG) is a new material for dam construction works, produced by adding cementitious materials and water to easily accessible rock based material, including sand and gravel at the river bed or excavation muck near the dam site, and mixing them with simple equipment and process. The cemented sand gravel dam (CSGD) built on CSG material has the advantages of both a roller compacted concrete dam (RCC dam) and a rockfill dam. Compared with an RCC dam, it demands less cement, with simplified aggregate preparation and mixing facilities. Moreover, temperature control measures are not necessary in the construction. As a result, it can effectively speed up the construction and lower the project cost. Compared with rockfill dams, a CSG dam requires a significantly lower amount of engineering works, but has batter capacity to withstand seepage deformation and scouring. Through the utilization of waste materials, CSG technology lowers the demand for artificial materials and high-quality aggregate, and thus promotes the efficient use of resources, less destruction of land vegetation, and a lower impact on the natural environment. Therefore, CSGD is a new kind of easily constructed dam that is economic, safe, low carbon, and eco-friendly.

In recent years, the relationship between dams and the natural environment has attracted increasingly public attention. It has become a trend of dam technology development to strike a balance between the low-cost and efficient construction of modern dams as well as a lower impact on the natural environment. As the basis and carrier of water resources and hydropower development, reservoir dams play a major role in the comprehensive utilization of water and hydropower resources. Their position will be further strengthened for the sustainable social and economic development of China, to which the CSGD development is the key technology.

According to statistics, dozens of CSGDs have been built around the world since the 1980s. Japan, Greece, Dominica, the Philippines, Pakistan, and Turkey are among countries that have already carried out engineering explorations and applied the technology [1–7]. China began the study of CSGD at the end of last century. By the beginning of this century, many colleges and research institutes have made extensive studies and discussions on the property, constitutive model, calculation and analysis methods, design principles, and standards of CSG material. The representative researchers include Jia Jinsheng, Zheng Cuiying et al. of China Institute of Water Resources and Hydropower Research [8–11], Sun Mingquan et al. of North China University of Water Resources and Electric Power [12–16], He Yunlong et al. of Wuhan University [17–20], and Cai et al. [21,22] of Hohai University. The previous studies believe that as CSG material only uses a small amount of cement, fly ash can be added in large quantities to improve its strength. However, the specific increased value of material strength and the optimal content of fly ash are not fully discussed. Therefore, the primary purpose of this study is to examine the influence of fly ash content on the compressive strength of CSG material. This quantitative examination provides various crucial insights for building dams of this type in the future.

## 2. Test Design

### 2.1. Test Materials

CSG material is composed of water, sand (fine aggregate), stone (coarse aggregate), and cementitious materials (cement and fly ash). The proportions of them can vary. In this test, (1) water: Zhengzhou tap water; (2) Sand: Natural river sand in Ruzhou County section of Ruhe River was adopted, the fineness modulus FM = 2.57, in accordance with the requirements of the Technical Guidelines for Damming with Cemented Granular Materials (SL 678-2014) that " . . . in natural materials, the fineness modulus of sand should range from 2.0 to 3.3". (3) Stone: to study the effect of aggregate gradation on the strength of CSG material, natural graded aggregates in Ruzhou County section of Ruhe River were used. After artificial screening, aggregates were divided into 5–20 mm, 20–40 mm, 40–80 mm, 80–150 mm and distributed in silos, as illustrated in Table 1. (4) Cementitious materials: the 425# ordinary Portland cement (P.O.42.5) was produced by Henan Duoyangda Cement Co., Ltd., Zhengzhou, China, which met the requirements for cement in the CSG material in the Technical Guidelines for Damming with Cemented Granular Materials (SL 678-2014) that "all Portland cement series conforming to GB175 and GB200 can be used to build dams with cemented granular materials. When mineral admixtures such as fly ash are added into cementitious materials, Portland cement, ordinary Portland cement, medium or low heat Portland cement should be preferentially selected." The dry discharged F Class II fly ash from Zhengzhou Thermal Power Plant was used in the test. The performance index is shown in Table 2.

**Table 1.** Aggregate gradation.

| Gradation | Coarse Aggregate/% | | | | | Fine Aggregate/% |
|---|---|---|---|---|---|---|
| | 5–20 mm | 20–40 mm | 40–80 mm | 80–150 mm | >150 mm | <5 mm |
| Proportion | 22.91 | 36.52 | 23.09 | 5.75 | 8.92 | 2.81 |

**Table 2.** Performance index of fly ash.

| Apparent Density/(g/cm$^3$) | 45 μm Sieving Residue (%) | Water Demand (%) | Chemical Composition (%) | | | | |
|---|---|---|---|---|---|---|---|
| | | | SiO$_2$ | Fe$_2$O$_3$ | Al$_2$O$_3$ | CaO | Loss of Burning |
| 2.11 | 17 | 102 | 59.61 | 7.41 | 21.33 | 4.24 | 1.78 |

### 2.2. Mix Proportion in the Test

In this test, "weight method" was used in a mixed proportion design. According to the Technical Guidelines for Damming with Cemented Granular Materials (SL 678-2014) and previous research [23–30], the apparent density of CSG material was selected as 2350 kg/m$^3$ (the apparent density was checked after the specimen test. The samples were all about 2350 kg/m$^3$, and the maximum fluctuation was within ±2%). According to the research results, factors such as water–binder ratio, sand ratio, and aggregate gradation have great influence on the material strength, and all factors have optimal values. This paper mainly focused on the influence of fly ash content on the strength of CSG material, so the proportion of other materials followed the reference of previous studies with the water–binder ratio at 1.0, sand ratio at 20%, coarse aggregate ranging from 20–40 mm, and 5–20 mm, mass ratio at 6:4, and cement content of 50 kg/m$^3$ and 60 kg/m$^3$. The fly ash content was designed to be 0 kg/m$^3$, 20 kg/m$^3$, 30 kg/m$^3$, 40 kg/m$^3$, 50 kg/m$^3$, 60 kg/m$^3$, 70 kg/m$^3$, 80 kg/m$^3$, 90 kg/m$^3$, and 100 kg/m$^3$, respectively as the admixture for cementitious materials. The designed mix proportion is shown in Table 3.

**Table 3.** Mix proportion of CSG material.

| Sample Code | Volume of Material Per Unit Volume/(kg/m$^3$) | | | | | |
|---|---|---|---|---|---|---|
| | C | F | W | S | NA1 (20–40 mm) | NA2 (5–20 mm) |
| C50F0 | | 0 | 50 | 450 | 1080 | 720 |
| C50F20 | | 20 | 70 | 442 | 1061 | 707 |
| C50F30 | | 30 | 80 | 438 | 1051 | 701 |
| C50F40 | | 40 | 90 | 434 | 1042 | 694 |
| C50F50 | 50 | 50 | 100 | 430 | 1032 | 688 |
| C50F60 | | 60 | 110 | 426 | 1022 | 682 |
| C50F70 | | 70 | 120 | 422 | 1013 | 675 |
| C50F80 | | 80 | 130 | 418 | 1003 | 669 |
| C50F90 | | 90 | 140 | 414 | 994 | 662 |
| C50F100 | | 100 | 150 | 410 | 984 | 656 |
| C60F0 | | 0 | 60 | 446 | 1070 | 714 |
| C60F20 | | 20 | 80 | 438 | 1051 | 701 |
| C60F30 | | 30 | 90 | 434 | 1042 | 694 |
| C60F40 | | 40 | 100 | 430 | 1032 | 688 |
| C60F50 | 60 | 50 | 110 | 426 | 1022 | 682 |
| C60F60 | | 60 | 120 | 422 | 1013 | 675 |
| C60F80 | | 80 | 140 | 414 | 994 | 662 |
| C60F100 | | 100 | 160 | 406 | 974 | 650 |

Note: Volume of material per unit volume/(kg/m$^3$); C, cement; F, fly ash; W, water demand; S, sand; NA1, coarse aggregate (20–40 mm); and NA2, coarse aggregate (5–20 mm).

### 2.3. Specimen Preparation

Considering that no standard CSG material test exists, and the composition of CSG is similar to concrete, this test followed the Test Code for Hydraulic Concrete (SL352-2006). As the construction and dam filling of CSGDs home and abroad have all adopted roller compaction, the CSG specimens in this paper were also formed through rolling compacted concrete, with the upper part of the specimens compacted and the lower part vibrated. After the vibration, the surface of specimens was smoothed and covered with a film (to prevent water evaporation). Before demolding, they were placed in a room with the

temperature of 20 °C ± 5 °C for 48 h [31–34]. After demolding, the specimens were sent to the standard maintenance room for maintenance lasting 28 d and 90 d. Eighteen groups were formed, with eight cubic specimens of 150 mm × 150 mm × 150 mm in each group. The preparation process is shown in Figure 1.

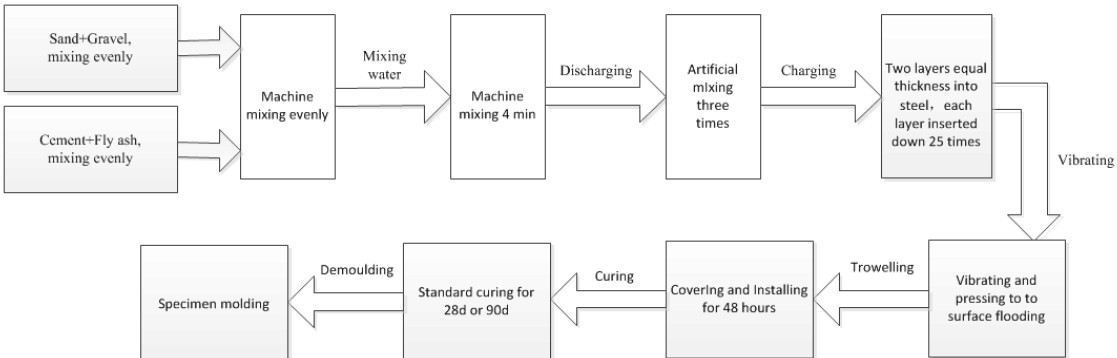

**Figure 1.** Flow chart of CSG sample preparation.

Preparation process of cemented sand and gravel material: Premixed stone and sand weighed according to test mix ratio, adding cement and fly ash, mixing evenly, and then adding water to mix, and finally mixing evenly and charging and vibrating. After the vibration is completed, the surface of the specimen is smoothed, and the film is covered (to prevent water evaporation). After standing at 20 °C ± 5 °C for 48 h, the demolded specimen is sent to the standard maintenance room for maintenance until the test age.

## 3. Test Results and Analysis

### 3.1. Determination of Compressive Strength

The cubic compressive strength of CSG material was followed the Test Rules for Hydraulic Concrete (SL352-2006).

In this test, four specimens were prepared as one group. After testing, the largest discrete data were removed and the other three groups were averaged. The test results were in accordance with the Rules.

### 3.2. Analysis of the Test Data

3.2.1. Effect of Fly Ash Content on the Compressive Strength of CSG Material

Fly ash is a type of artificial pozzolanic material made of silica or silica-alumina. It possesses minimal or no cementing value. However, in the presence of water, fly ash powder will react with $Ca(OH)_2$ at room temperature to form a compound with cementing properties. The pozzolanic activity of fly ash is reflected by its cementing.

The test results are shown in Figure 2 for the analysis of the relationship between age, fly ash content, and strength.

As detailed in Figure 2, the rules can be found as follows: (1) the compressive strength of CSG increases with age. The compressive strength of CSG at 90 d is generally 10–30% higher than that at 28 d. (2) When the cement content is 50 kg/m$^3$ and 60 kg/m$^3$, respectively, the compressive strength of CSG increases with the addition of fly ash content. When the cement content is 50 kg/m$^3$, the increase of compressive strength at 28 d is greater than that at 90 d. When the cement content is 60 kg/m$^3$, the increase of compressive strength at 90 d is greater than that at 28 d [35].

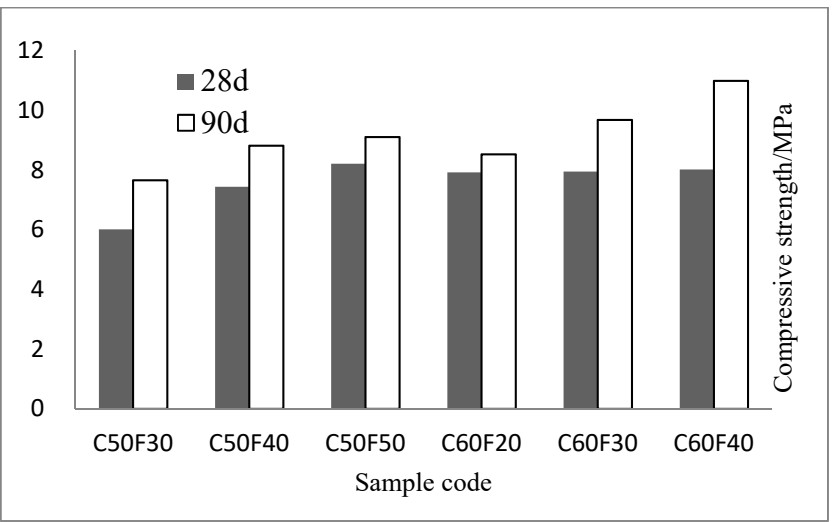

**Figure 2.** The relationship between the compressive strength of CSG material and the age/the content of fly ash.

The test results can be explained with two-phase hydration reaction of the fly ash-cement system. First, the induction stage, during which the soluble ions on the surface of fly ash particles dissolve, which affects the nucleation of $Ca(OH)_2$ and C-S-H hydrates and retards the initial level of $C_3A$ in cement. Meanwhile, the hydration of cement is delayed, as the fine size of fly ash particles leads to their easy adhesion to the surface of cement particles under physical action. In addition, $Ca(OH)_2$ and C-S-H produced by cement hydration wrap the surface of fly ash and prevent the hydration of fly ash particles. Therefore, the fly ash with early age has low self-activity. Second, the acceleration stage, during which $Ca(OH)_2$ and C-S-H begin to nucleate and grow. Fly ash particles provide additional accommodation for the precipitation of C-S phase hydrates in cement through many new surfaces. In this way, fewer C-S phase hydrates will precipitate on the surface of cement particles, and the dissolution of $C_3S$ will accelerate. As a result, the hydration of cement is promoted. Therefore, during the second phase, the presence of fly ash facilitates the hydration of cement. Moreover, due to the nucleation of cement hydration products, the permeability of fly ash coating increases and leads to faster hydration of the fly ash. As fly as hydration consumes more $Ca(OH)_2$, the hydration of cement is promoted further. During this stage, cement and fly ash promote and accelerate each other's hydration at the same time. This phenomenon become obvious with the development of age, which is highlighted by the significant increase of strength of CSG material at the later stage. Therefore, considering the gradual development of material strength, it is appropriate to choose 90 d or 180 d strength as the design strength of CSG dam.

3.2.2. Optimal Fly Ash Content of CSG Material

Most hydraulic projects will add fly ash to cementitious materials, and the RCC dam has the highest proportion of fly ash addition (70%) (Jiangya Hydropower Station). Technical Guidelines for Damming with Cemented Granular Materials (SL678-2014) also proposes including fly ash into cement gravel and sand material for dam construction, but without mentioning the most optimal and economical fly ash content.

During the test, prepared the cement content of 50 kg/m$^3$ and 60 kg/m$^3$, added fly ash of 20 kg/m$^3$, 30 kg/m$^3$, 40 kg/m$^3$, 50 kg/m$^3$, 60 kg/m$^3$, 80 kg/m$^3$, or 100 kg/m$^3$, respectively, into the cement and waited for 90 d maintenance to make CSG materials to see the difference of their compressive strengths, with results drawn in the figure below. According to the figure, as the fly ash content increases, the compressive strength reaches a peak before falling down to a level. The CSG material test proves that an optimal content of fly ash could be found. The compressive strength of materials will reach its maximum

when both cement and fly ash contents stand at 50 kg/m$^3$ or 60 kg/m$^3$, as highlighted in Figure 3.

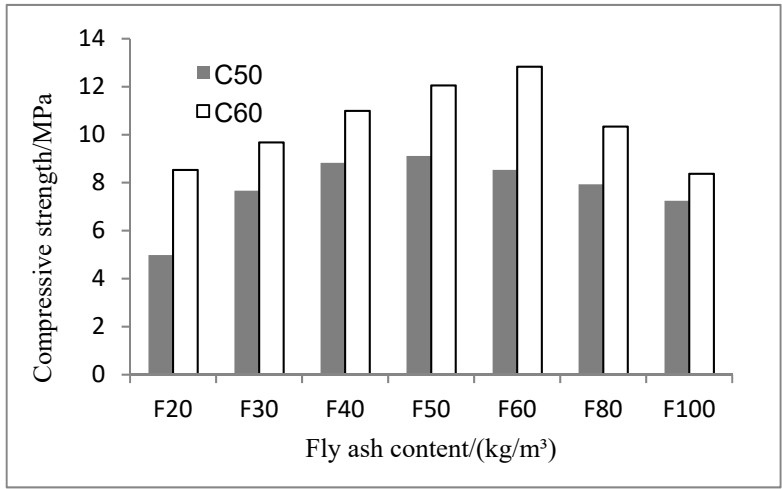

**Figure 3.** Optimal content of fly ash of CSG material.

According to our analysis, as the design of mixed proportions in this test is based on previous research results, the total water consumption of the mixture is fixed by selecting a water–binder ratio of 1.0 and sand ratio of 20%. [35,36] At the initial stage, the strength of CSG material increases with the increase of fly ash content and the water amount is enough to support the hydration of cement and fly ash. The increase of fly ash facilitates C-S-H cementing; when all the water is consumed during the hydration of cement and fly ash, the fly ash added will only serve as a filler, which fill in the concrete voids or attach to the surface of cement particles. At this stage, more fly ash will affect the formation of Ca(OH)$_2$, C-S-H, which leads to the declining strength of CSG materials. Therefore, there is an optimal fly ash content in CSG material, and it is defined to be 50% of the contentious material (cement + fly ash).

## 4. Numerical Regression Analysis

Regression analysis is an important branch of mathematical statistics and an important statistical tool to study the correlation between variables. It is well applied in many areas, such as seeking empirical formulas or establishing mathematical models.

According to the test, fly ash content has significant influence on the compressive strength of CSG material. Based on the theory of linear regression model, Formula (1) is built to reflect the statistical relationship between 90 d compressive strength and fly ash content in CSG material:

$$X_F = a_f x_f^2 + b_f x_f + c_f \tag{1}$$

where $a_f$, $b_f$, and $c_f$ are statistical constants.

After calculation, the regression equation parameters of fly ash content and compressive strength are obtained and shown in Table 4.

As illustrated in Table 4, the fitting correlation coefficients are greater than 0.90 and the standard errors are less than 0.58 in all cases. The fitting strength of each regression equation and measured values have demonstrated excellent correlation with regard to 90 d compressive strength of CSG material, with small deviation of regression values and measured values, and high fitting accuracy. The comprehensive analysis shows that the test results suit the mathematical model of linear regression.

Through numerical analysis, the maximum value of compressive strength under different fly ash content can be obtained, where x represents the optimal fly ash content and y represents the maximum compressive strength of CSG material, as detailed in Table 4.

The numerical analysis shows that when (1) 50 kg/m$^3$ of cement content is combined with 51 kg/m$^3$ of fly ash content, or (2) the cement and fly ash contents are 60 kg/m$^3$ and 59 kg/m$^3$, respectively, the compressive strength of CSG material is the highest in 90 d. Therefore, when fly ash content: cement content ≈ 1:1 or when the fly ash content is 50% of cementitious materials (cement + fly ash), the "optimal content" is achieved. The "optimal fly ash content" of CSG material was verified again through numerical analysis.

**Table 4.** Regression analysis parameter table.

| Cement/ (kg/m$^3$) | Fly Ash/(kg/m$^3$) | Compressive Strength/(MPa) | Fitting Strength/ (MPa) | Absolute Error | Correlation Coefficient | Standard Error | $a_f$ | $b_f$ | $c_f$ | Optimal Fly Ash Content/ (kg/m$^3$) | Corresponding Strength/ (MPa) |
|---|---|---|---|---|---|---|---|---|---|---|---|
| | 30 | 7.94 | 8.236 | −0.3 | | | | | | | |
| | 40 | 8.82 | 8.563 | 0.26 | | | | | | | |
| | 50 | 9.11 | 8.726 | 0.38 | | | | | | | |
| 50 | 60 | 8.53 | 8.726 | −0.2 | 0.9 | 0.39 | −0.00082 | 0.09 | 6.273 | 51 | 8.22 |
| | 80 | 7.93 | 8.233 | −0.3 | | | | | | | |
| | 100 | 7.24 | 7.086 | 0.15 | | | | | | | |
| | 20 | 8.53 | 8.35 | 0.18 | | | | | | | |
| | 30 | 9.68 | 10.015 | −0.34 | | | | | | | |
| | 40 | 10.99 | 11.198 | −0.21 | | | | | | | |
| 60 | 50 | 12.05 | 11.896 | 0.15 | 0.96 | 0.58 | −0.0024 | 0.288 | 3.567 | 59 | 12.11 |
| | 60 | 12.83 | 12.111 | 0.72 | | | | | | | |
| | 80 | 10.34 | 11.089 | −0.75 | | | | | | | |
| | 100 | 8.37 | 8.132 | 0.24 | | | | | | | |

## 5. Conclusions

As a new type of eco-friendly material for dam construction, CSG is a low-cost material as it requires a lower amount of cement content and higher fly ash content than traditional ones. This paper briefly introduced the steps of CSG material test. The influence of fly ash content on the strength of CSG material was studied from two perspectives—test work and numerical analysis. The conclusions are as follows:

(1) The compressive strength of CSG increases with age. The compressive strength of CSG at 90 d is generally higher by 10–30% compared with that at 28 d. After taking into account the gradual development of material strength, 90 d or 180 d strength is recommended as the design strength for the CSG dam.

(2) Test work and numerical analysis have verified that an optimal fly ash content for CSG material exists, and is defined as 50% of the total cementitious material (cement + fly ash). The results are of material significance in the solution of problems during the design and construction of CSG dam.

(3) In the new material of CSG, the gravel material of natural river is selected to study the tensile, compressive, and shear tests under different particle gradation, sand rate, water–cement ratio, and cementitious material dosage, and the tensile, compressive, and shear strength indexes are obtained. This paper mainly takes the content of fly ash as an example to study the compressive strength characteristics of materials. In the follow-up study, the mechanical properties of materials such as compressive strength, tensile strength, and shear strength under multiple factors will be closely analyzed.

**Author Contributions:** All authors contributed to the study conception and design. Q.C. performed all experiments and analysis along with data collection and discussion of the results. F.W. (Fang Wan) wrote the manuscript. L.X. edited the manuscript. F.W. (Feng Wu) reviewed the manuscript. All authors have read and agreed to the published version of the manuscript.

**Funding:** The research was supported by Central Plains Science and Technology Innovation Leading Talent Funding Project (194200510008); Major Science and Technology Special Projects in Henan Province (201300311400).

**Data Availability Statement:** The data used to support the findings of this study are included within the article.

**Conflicts of Interest:** We declare that we have no conflict of interest or the authors do not have any possible conflicts of interest.

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
