# Peer review of "The Influence of Fly Ash Content on the Compressive Strength of Cemented Sand and Gravel Material"

_crystals, doi:10.3390/cryst11111426_

Round 1

Reviewer 1 Report

The introduction has little to do with the title of the article and the purpose of the work. There is a lack of overview of the problem of CSG material strength, peculiarities of the hydration, microstructure formation of such materials and etc.). 

An important characteristic of CSG material was determined in the work - compressive strength, depending on the composition of the material. However, there is a lack of other studies to explain the strength results obtained at work. These studies are related to the evaluation of the hydration and structure formation of the material. Therefore, I suggest that the interpretations of the results (page 6 and 7) should be based on the citation of other works.

One more remark. The introduction notes that the CSGD has been researched and developed since 1990s. So, it is not clear why the authors call CSG as new material (Introduction), new type of eco-friendly material (Conclusions).

Author Response

Please see word

Reviewer 2 Report

The review is in the attachment.

Author Response

Please see word

Round 2

Reviewer 1 Report

the corrections and explanations provided by the authors have been accepted by me, so I have no comments

Author Response

Thank you for your comments!

Reviewer 2 Report

The paper can be accepted for publication.

Author Response

Thank you for your comments!